# Pelvic Lymphadenectomy in Gynecologic Oncology—Significance of Anatomical Variations

**DOI:** 10.3390/diagnostics11010089

**Published:** 2021-01-07

**Authors:** Stoyan Kostov, Yavor Kornovski, Stanislav Slavchev, Yonka Ivanova, Deyan Dzhenkov, Nikolay Dimitrov, Angel Yordanov

**Affiliations:** 1Department of Gynecology, Medical University Varna “Prof. Dr. Paraskev Stoyanov”, 9002 Varna, Bulgaria; drstoqn.kostov@gmail.com (S.K.); ykornovski@abv.bg (Y.K.); st_slavchev@abv.bg (S.S.); yonka.ivanova@abv.bg (Y.I.); 2Department of General and Clinical Pathology, Forensic Medicine and Deontology, Division of General and Clinical Pathology, Faculty of Medicine, Medical University Varna “Prof. Dr. Paraskev Stoyanov”, 9002 Varna, Bulgaria; dzhenkov@mail.bg; 3Department of Anatomy, Faculty of Medicine, Trakia University, 6000 Stara Zagora, Bulgaria; nikolay.dimitrov@trakia-uni.bg; 4Department of Gynecologic Oncology, Medical University Pleven, 5800 Pleven, Bulgaria

**Keywords:** anatomical landmarks, anatomical variations, pelvic lymph nodes, gynecologic oncology, pelvic lymphadenectomy

## Abstract

Pelvic lymphadenectomy is a common surgical procedure in gynecologic oncology. Pelvic lymph node dissection is performed for all types of gynecological malignancies to evaluate the extent of a disease and facilitate further treatment planning. Most studies examine the lymphatic spread, the prognostic, and therapeutic significance of the lymph nodes. However, there are very few studies describing the possible surgical approaches and the anatomical variations. Moreover, a correlation between anatomical variations and lymphadenectomy in the pelvic region has never been discussed in medical literature. The present article aims to expand the limited knowledge of the anatomical variations in the pelvis. Anatomical variations of the ureters, pelvic vessels, and nerves and their significance to pelvic lymphadenectomy are summarized, explained, and illustrated. Surgeons should be familiar with pelvic anatomy and its variations to safely perform a pelvic lymphadenectomy. Learning the proper lymphadenectomy technique relating to anatomical landmarks and variations may decrease morbidity and mortality. Furthermore, accurate description and analysis of the majority of pelvic anatomical variations may impact not only gynecological surgery, but also spinal surgery, urology, and orthopedics.

## 1. Introduction

Pelvic lymph node dissection (PLND) is a common surgical procedure in gynecologic oncology [1]. The lymphatic system is the primary dissemination pathway for gynecological malignancies. PLND is applied for cancer staging, prognosis, surgical, and postoperative management [2,3]. PLND is performed for all types of gynecological malignancies to evaluate the extent of a disease and facilitate further treatment planning. Additionally, PLND is beneficial in cases where removing metastatic lymph nodes improves overall survival and disease-free survival [4]. Most studies examine the lymphatic spread, the prognostic, and therapeutic significance of pelvic lymph nodes. However, there are very few studies describing the possible surgical approach, dissection techniques and anatomical variations [5]. There is limited information and disagreement on lymph nodes location, groups, and overall number [6].

Moreover, a correlation between anatomical variations and PLND in the pelvic region has never been discussed in medical literature. Surgeons should be familiar with pelvic anatomy and its variations to safely perform PLND. Learning the proper lymphadenectomy technique relating to anatomical landmarks and variations may decrease morbidity and mortality [7]. The present article aims to define, detail, and summarize the anatomic landmarks during PLND in gynecologic oncology. Furthermore, a summary of the most common anatomical variations (of nerves, vessels, ureters) and potential complications related to PLND in the pelvic region are clearly defined.

## 2. Pelvic Lymph Nodes and Regions

Knowledge of the anatomical localization of lymph node groups in the pelvis is essential for the effectiveness and safety of lymphadenectomy [8]. The pelvic lymph nodes and the connecting lymphatic channels communicate with the venous system; the lymphatic system embryologically develops from vascular plexuses, arising from the venous system [3]. The anatomical localization of the major groups and sub-groups of pelvic lymph nodes is summarized in Figure 1, Figure 2 and Figure 3 [2,6,8,9,10,11,12,13,14,15].

Some authors include the subaortic and promontory lymph nodes in the medial common iliac lymph group; the obturator lymph nodes may also be considered as part of the medial external lymph group. Moreover, in some articles, anterior, lateral sacral, and gluteal internal lymph nodes are defined as junctional lymph nodes [8,10,12].

We present an anatomical classification of the pelvic lymph nodes rather than a clinical one. Although the anatomical classification is rather complex, it better defines the localization of the pelvic lymph nodes. Furthermore, the anatomical variations, significant for the PLND, are better described.

In medical literature and among surgeons, there are large variations and discrepancies in the nomenclature of the pelvic lymph node regions [2]. The present article and the majority of authors recognize the following pelvic lymph node regions: common iliac; external iliac; internal iliac; obturator; and sacral (or presacral) [5,7,12,13]. Some authors add parametrial, mesorectal, and interiliac regions, while others include tissue from the interiliac region to the external iliac and obturator regions and remove the tissue from the parametrial region together with the parametrium during radical hysterectomy [5,14,15].

## 3. Selective PLND—Anatomical Landmarks and Techniques

There are several surgical procedures related to dissection of pelvic lymph nodes: sentinel lymph node biopsy, pelvic lymph node sampling, selective lymphadenectomy and complete (systematic) PLND. The present article describes the systematic lymphadenectomy in which all pelvic lymph nodes, draining the pelvic organs, have been removed [1]. Cibula and Rustum presented detailed anatomic boundaries for five pelvic lymph node regions: external iliac, obturator, internal iliac, common iliac, and presacral region [5]. Anatomic boundaries of the common PLND are: ventral—common iliac artery bifurcations, dorsal—abdominal aorta bifurcation, lateral—psoas major muscle, medial—right-medial aspect of common iliac vessels, left-mesoureter [5]. Anatomic boundaries of the external and the internal PLND include common iliac artery bifurcation dorsally, the deep circumflex iliac vein ventrally, and the genitofemoral nerve of the psoas major muscle laterally. The medial border is the obliterated umbilical artery ventrally and the ureter dorsally (Figure 4 and Figure 5) [1,7,15].

## 4. Systematic Open PLND—Surgical Technique and Steps

(1)Peritoneal incision. After transecting (not a necessary step) the round ligament, the peritoneum is incised in a posterior (lateral and parallel to the infundibulopelvic ligament) and an anterior (ventrally and laterally to the obliterated umbilical artery) direction. The iliac vessels are exposed from the bifurcation of the aorta to the inguinal ligament.(2)Identification of the ureter.(3)Lateral paravesical and lateral (Latzko’s space) pararectal space dissection. The lateral paravesical space is dissected between the obliterated umbilical artery and the external iliac vessels. Lateral pararectal space is dissected between the internal iliac artery and the ureter.(4)Genitofemoral nerve identification. Lateral incision to the fascia of the psoas muscle is preferable in order to avoid genitofemoral nerve injury. The nerve is located lateral to the external iliac vessels and sometimes overlying them.(5)External iliac region dissection: lateral and medial external iliac vessels dissection. The dissection begins at the origin of the external iliac vessels and finishes down to the point where the deep circumflex iliac vein crosses over the external iliac artery.(6)Obturator region dissection. The obturator space is approached by retracting the external iliac vessels medially and the psoas muscle laterally, and by dissection of the areolar tissue that lies directly between these vessels and the lateral pelvic wall. The obturator nerve is identified. The procedure is followed by lateral retraction of the external iliac vessels to expose the obturator space. Superficial obturator lymph nodes are dissected after obturator nerve visualization (obturator nerve stripping). For locally advanced cervical cancer cases, PLND continues with dissection of the deep obturator lymph nodes and the gluteal nodes.(7)Internal iliac region dissection. Lymph nodes are removed medially and anteriorly to the internal iliac vessels.(8)Common iliac region dissection. Lymph nodes are removed ventrally and laterally from both common iliac vessels to the aortic bifurcation. Middle common iliac lymph nodes are located in the lumbosacral fossa. It is approached by medial retraction of the common iliac vessels and lateral retraction of the psoas muscle. The obturator nerve, entering the obturator fossa through the body of the psoas muscle, the iliolumbar artery/vein, and the lumbosacral plexus are exposed.(9)Sacral (presacral) region dissection. After medial mobilization of the sigma-rectum, the peritoneum and the presacral fascia are incised medially to the right common iliac artery. Sacral lymph nodes, localized below the bifurcation of the abdominal aorta and inferior vena cava, in the triangle between the left and right common iliac vessels, are dissected [1,5,7,16,17,18,19].

## 5. Anatomical Landmarks and Anatomical Variations, Related to Systematic PLND

Anatomy of the pelvic ureter. Identification of the ureter is a necessary step during PLND for two reasons: to avoid injury and to serve as a medial landmark during PLND. The ureter is divided into abdominal, pelvic, and intramural segments [20]. According to Luschka’s law, the left ureter crosses the iliac artery 1.5 cm below the common iliac artery bifurcation, while the right ureter crosses the iliac artery 1.5 cm above the bifurcation. Therefore, the ureter enters the pelvic cavity by crossing the common iliac artery on the left side and the external iliac artery on the right [21,22]. Hence, the left ureter is located laterally to the internal iliac artery, whereas the right ureter is located medially to the right internal iliac artery. Surgeons should fully understand the anatomical relation between the ureter and iliac vessels. As the ureter enters the true pelvis, it runs caudally and medially to the ovarian vessels, reaching the bladder on the posterior leaf of the broad ligament [23].

## 6. Anatomical Variations of the Ureter, Related to PLND in Gynecologic Oncology (PLNDGO)

There is a multitude of ureteral anomalies, but the following three groups are related to PLND in gynecologic oncology (PLNDGO):(A)Multiplication of ureter;(B)Ureteral diverticulum;(C)Unusual ureteral position—retro-iliac ureter.

(A) Ureteral multiplication is a longitudinal segmentation of the ureter into two or more tubes. Multiplication may be complete, incomplete, bilateral, and unilateral (Figure 6). Ureteral duplication is the most common type of multiplication and occurs more commonly in women than in men (4–5% of the population). Incomplete duplications tend to be unilateral and more common than complete, whereas complete duplications are often bilateral [24,25,26]. Unilateral ureteral duplication is more common on the right ureter than on the left (Figure 6) [25].

Surgical considerations. Ureteral multiplication increases the incidence of ureteral injury during PLND. Surgeons have to identify the course of the ureter as it crosses the pelvic brim and reaches the bladder. Injuries to the blood vessels of the ureters must be avoided, as duplicated ureters are usually contained within a single sheath and the associated blood supply could be interrupted [24,26]. Additionally, one of the duplicated ureters might be confused with vessel structures (artery most frequently), cut, and ligated during dissection. If intraoperatively, a suspicion of ureteral duplication arises, surgeons should observe the peristaltic activity of both ureters to differentiate this variation.

(B) Ureteral diverticulum is a rare urological congenital anomaly, classified into three subgroups: (1) abortive ureteral duplications, sharing the same embryogenesis as diverticula of the disordered ureteral budding; (2) true (congenital) diverticulum, characterized by the presence of all tissue layers of the normal ureter; and (3) false (required) diverticulum, which represents a mucosal herniation [27,28]. Ureteral diverticula are mainly asymptomatic, although urinary tract infection and ureteral stones causing obstruction could appear. Papin and Eisendrath proposed and illustrated urethral diverticulum classification, which can be used clinically for PLNDGO (Figure 6) [29].

Surgical considerations. Ureteral diverticulum may also be confused with vascular structures. Ampullary diverticulum could mimic retroperitoneal cysts. In cases of retroperitoneal tumors in the pelvic region during PLNDGO, the ureter must be identified.

(C) Retro-iliac ureter (RIU) is a rare urological condition in which the ureter passes deep to the iliac vessels. Generally, the condition is diagnosed intraoperatively. Despite being a congenital anomaly, for the majority of patients the RIU first manifests itself as flank pain and symptoms of ureteral obstruction in their second or third decade of life. Coexisting anomalies are common: vaginal atresia, lumbosacral agenesis, or malrotation of the kidney. Surgical treatment consists of a dissection of the ureter and its anterior repositioning with a subsequent reimplantation in the bladder wall.

RIU is thought to originate as a result of embryologic defects: a defect in the mesonephric ureteral migration during arterial development, persistence of the embryologic primitive ventral root of the umbilical artery, between the aorta and distal umbilical artery, traps the ureter dorsally. The third hypothesis involves abnormal development of the iliac vessels from the anterior branch of the umbilical artery instead of the dorsal [30,31,32]. RIU is commonly located behind the common iliac artery, but other previously reported types have been described as a retro-common iliac vein, retro-external iliac artery/vein, and retro-internal iliac artery [30,31,32]. A bilateral RIU is a rare but possible variation (Figure 6) [30].

Surgical considerations. Although a rare anatomic variation, the RIU is of high importance. First, it is hard to be identified if the surgeon is not familiar with such a variation of the ureter. Second, it could be injured during an iliac lymph nodes dissection, especially if it is mistaken for a vascular structure. Finally, the RIU could not be the dorsomedial border of the PLND, as it has different location. In such cases, the lateral aspect of the rectum is considered as the dorsomedial border.

Anatomical variations of the ureter—conclusion of surgical considerations.

Ureteral injuries (UIs) occur approximately in 0.5–1% of all pelvic operations with gynecological operations accounting for 75% of these. UIs are more common during radical hysterectomy procedures combine with PLND. The incidence of UIs during PLND is 10%. It is believed that ureteral variations lead to an increased risk of UIs. There is consensus that a multiplication of the ureter is an independent risk factor for UIs during PLND. Benedetti-Panici et al. reported a prospective study, which included 309 consecutive patients with cervical, endometrial, and ovarian cancer treated with systematic aortic and PLND. Ureteral duplication was observed in four (1.6%) cases. Authors concluded that knowledge of ureteral anomalies is important in gynecological surgery, as the risk of UIs is elevated. Duplicated UIs during gynecological procedures are reported in medical literature. UIs may occur at the level of the infundibulopelvic ligament and deep in the pelvis, below the level of ischial spine, where the ureter lies lateral to the peritoneum of uterosacral ligament. Anatomical variations of the ureter are often associated with other congenital anomalies. Patients with known congenital anomalies or different types of syndromes should undergo preoperative imaging for careful surgical planning [33,34,35,36,37,38,39]. Postoperatively, a routine cystoscopy should be performed to rule out UIs [26].

**Figure 6 diagnostics-11-00089-f006:**
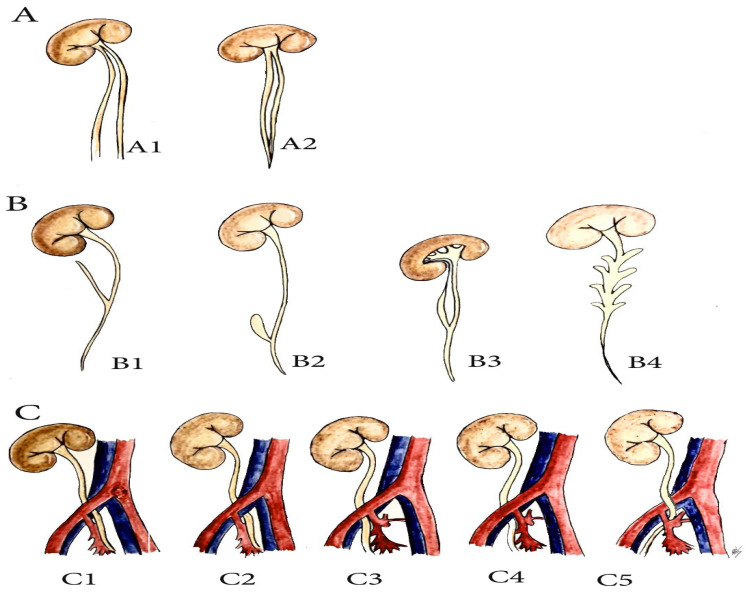
Anatomical variations of the ureter related to pelvic lymph node dissection in gynecologic oncology (PLNDGO). (**A**) Duplicated ureter. (**A1**) complete duplication. (**A2**) incomplete duplication. (**B**) Ureteral diverticulum (Adapted from Papin and Eisendrath [29]). (**B1**) simple diverticulum, (**B2**) ampullary diverticulum, (**B3**) diverticulum ending in fibrous prolongation, (**B4**) multiple diverticulus. (**C**) Retroiliac ureter. (**C1**) Behind the common iliac artery, (**C2**) behind the common iliac vein, (**C3**) behind the external iliac artery, (**C4**) behind the external iliac vein, (**C5**) behind the internal iliac artery.

## 7. Iliac Vessel Variations

### 7.1. Iliac Arteries

As PLND is mainly a vascular dissection procedure, the anatomical variations of iliac vessels should be respected in order to avoid unnecessary hemorrhage or transfusion [40].

### 7.2. Common Iliac Artery (CIA) Anatomy

The abdominal aorta bifurcates anterolaterally to the left side of the fourth lumbar vertebral body and divides into the left and right common iliac arteries. They further divide into the external and internal iliac arteries at the level of the sacroiliac joint. The right CIA (5 cm) is frequently longer than its left counterpart (4 cm) [41].

### 7.3. CIA Variations Related to PLNDGO

CIA variations are relatively rare. Although the true incidence of CIA variations is not known, the most frequent anomaly is an absence of the CIA. It tends to be unilateral, predominantly to the right, even though bilateral agenesis of CIA has also been described [42,43,44]. In one of the cases, described by Shetty et al., the abdominal aorta was directly branching into external and internal iliac arteries at the level of the 4th–5th lumbar vertebra [43]. Dabydeen et al. described a case of congenital absence of the right CIA. In their article, the proximal part of the right external iliac artery was absent. The distal part and the right common femoral artery was reconstituted from the right inferior epigastric artery, deep circumflex iliac artery, and the contralateral common femoral artery [44]. Llauger et al. presented a case of atresia of the right CIA, associated with a large aberrant and anomalous artery, connecting both hypogastric arteries within the pelvis [45].

Rusu et al. observed the anomalous origin of the iliolumbar artery, which has not been reported before. The authors dissected 15 human adult cadavers (30 pelvic halves). The CIA appeared trifurcated due to a higher origin of the iliolumbar artery (originating from the CIA bifurcation) in 2.5% of cadavers [46]. In 8.75% of specimens, the iliolumbar artery originated from the CIA [46].

Surgical considerations. In cases of absent CIA, surgeons should be aware of collateral, anomalous, or aberrant arteries. Collateral arteries serve as an alternative blood source, compensating the absent CIA; they might be of atypical origin from the iliac system. Iliolumbar arteries, originating from the CIA or CIA bifurcation, may be damaged during a middle common iliac lymph nodes dissection of the lumbosacral fossa.

## 8. External/Internal Iliac Artery Anatomy

The external iliac artery (EIA) is a direct continuation of the CIA. It runs downwards, forwards in the iliac fossa, and reaches the lacuna vasorum under the inguinal ligament. The EIA has two branches: inferior epigastric artery and deep circumflex iliac artery. After crossing the mid-inguinal point, it continues as a common femoral artery. The internal iliac artery (IIA) arises from the CIA anterior to the sacroiliac joint. In the majority of the cases, the artery originates at the level of the L5–S1 intervertebral disc. The IIA descends posteriorly towards the superior border of the greater sciatic foramen where it divides into two branches: anterior and posterior [42,47,48]. The anterior and posterior branches of the IIA are illustrated in Figure 7.

## 9. EIA Variations Related to PLNDGO

Variations of the EIA are grouped into four categories [48,49]:(A)Hypoplasia, agenesis of the EIA;(B)Anomalous origin and position of the EIA;(C)Morphological variations (variations in the shape of the EIA);(D)Variations in the branching pattern of the EIA.

(A) Kawashima et al. reported a rare case of hypoplastic right EIA, which continued into the normal femoral artery by anastomoses formed with the enlarged obturator and deep circumflex iliac arteries [50]. Okamoto et al. reported a case of the CIA entering into the pelvic cavity without branching to the EIA. The artery passed behind the first sacral nerve and gave rise to each branch of the IIA in the pelvis [51]. In cases of EIA hypoplasia, an ipsilateral persistent sciatic artery may also be observed [49].

Surgical considerations. Cases of EIA hypoplasia and agenesis are of clinical significance due to the possible presence of a collateral pathway. PLND should be meticulous in order to preserve collateral and anomalous vessels.

(B) The EIA may directly originate from the abdominal aorta. The origin of the EIA could be posterior in the internal iliac fossa. The position of the EIA may differ to the external iliac vein (EIV) position. The EIA might be located superficially or medially to the EIV [48,49].

Surgical considerations. During an external iliac lymph nodes dissection, surgeons should be familiar with the EIA position differences to prevent iliac vessel injury. Safi et al. reported a case of EIA laparoscopic injury due to an anomalous position of the artery. The EIA was elongated and located posterior and medial to the EIV. They repaired the artery using a two-needle single-knot technique and continued the lymphadenectomy. Authors concluded that during PLND, both external iliac vessels should be exposed and visible to the surgeon, as the anatomical variations of the EIA increased the risk of injury [52].

(C) Morphological variations of the EIA are grossly classified in five groups: looped, tortuous, curved, twisted, and S-shaped EIA. Nayak et al. presented a cadaveric study of 48 hemipelvises. They observed morphological variations in nine (19%) of the hemipelvises [48]. Looped and tortuous EIA have been reported in other case studies [53,54]. We also observed morphological variations of EIA (Figure 8).

Surgical considerations. Knowledge of the EIA morphological variations is essential during an external iliac lymph nodes dissection. Particular attention should be paid during dissection of the lateral, middle, and medial external iliac lymph nodes. The risk of iatrogenic EIA lesions may increase as a result of existing morphological differences.

(D) Variability in the origin of the obturator artery (OA) from the EIA has also been reported in medical literature [55,56]. If the OA originates from the external iliac system, it crosses the superior pubic ramus and the external iliac vein vertically [55]. The OA can also arise as a common trunk with the inferior epigastric artery and the deep circumflex iliac artery (Figure 9) [55,56]. Nayak et al. reported duplication of the deep circumflex iliac artery and an additional large muscular branch, arising and descending laterally from the EIA [48]. The medial circumflex femoral artery and the profunda femoral artery could also arise from the EIA [42].

Surgical considerations. An OA, arising from the EIA, is not a rare variation. The majority of OA origin variations are located at the distal part of the EIA.

## 10. IIA Variations, Related to PLNDGO

Most of the IIA variations are related to its branching pattern. Different authors have suggested various concepts for classification of the ending branches of the IIA. Most researchers and clinicians refer to the Adachi classification, which has been the standard for many years. Adachi classified the branching of the IIA using its four large parietal branches: the umbilical, superior gluteal, inferior gluteal, and the internal pudendal arteries (Figure 10) [42,57,58,59].

Surgical consideration. As shown in Figure 10, the SGA is a variable artery. Surgeons should be aware of the SGA variations to safely perform gluteal lymph node group dissection.

## 11. Uterine and Obturator Artery Variations Related to PLNDGO

Uterine artery. In the majority of cases, the uterine artery (UA) arises from the anterior branch of the IIA. Different origins of the UA have been reported: from the umbilical artery as a separate branch and via a common trunk (Figure 11); from the inferior gluteal artery as a separate branch; as a bifurcation from the inferior gluteal; and as a trifurcation with the superior and the inferior gluteal arteries [60].

Surgical considerations. The risk of iatrogenic injury to the UA is increased when the artery arises from the umbilical artery, as the UA crosses the operative field. If the PLND is performed before hysterectomy, surgeons should first identify the UA origin.

Obturator artery. The OA, branching from the EIA, has been discussed above. Herein, most of the OA anomalies will be analyzed.

In most cases, the OA originates from the anterior trunk of the IIA [55]. The OA runs anteroinferiorly and lies longitudinally to the obturator foramen on the medial part of the obturator internal muscle. It gives several branches within the pelvis, before entering the obturator foramen. They are classified as iliac, vesical, and pubic branches. The OA is located medially to the ureter, cranially to the obturator vein, and caudally to the obturator nerve [55,61]. The OA has the greatest variation frequency among the IIA branches [62]. Furthermore, the OA may arise from the EIA, the femoral artery, the deep circumflex iliac artery, the posterior branch of IIA, or from the inferior epigastric artery (Figure 12) [42,55,63]. In 20% to 34% of the cases, the pubic branch of the inferior epigastric artery replaces the OA. In such cases, the OA passes posterior to the lacunar ligament and courses into the superior pubic ramus vertically to enter the obturator foramen [55]. An OA arising from the EIA or its branches is classified as an aberrant obturator artery. Two obturator arteries could be observed during dissection. An additional OA with a different origin or path through the obturator fossa is classified as an accessory obturator artery.

Surgical consideration. Surgeons should be aware of the possible presence of accessory or aberrant obturator arteries during external and internal iliac lymph node groups dissection. The presence of normal OA does not exclude the existence of an accessory OA. The inferior epigastric artery is located below the deep circumflex iliac vein. Any injury of the OA arising from the inferior epigastric artery should be avoided, as the deep circumflex iliac vein is the ventral border of the dissection.

Anatomical variations of iliac arteries—conclusion of surgical considerations.

Arterial injuries during PLNDGO are less common than venous ones. There are very few studies describing arterial damage during PLNDGO. Bae et al. presented a retrospective review study. Authors observed four (1.3%) cases of major vascular injuries during 225 laparoscopic lymph node dissections. One of the injuries was to the EIA. Ricciardi et al. and Ishikawa et al. also reported damage to the EIA in a course of PLND. These studies showed that the EIA is more likely to be injured during PLND than the other iliac arteries. Gyimadu et al. concluded that the occurrence of vascular injuries during the course of PLNDGO may be due to variations in the anatomy of great retroperitoneal vessels. In medical literature the frequency of such abnormalities varies between 5.6 and 23.0% [64,65,66,67,68].

## 12. Iliac Veins

### 12.1. Common Iliac Vein Anatomy (CIV)

Common iliac veins (CIVs) drain into the inferior vena cava. They are a direct continuation of the external iliac veins and are formed by the junction of the external and internal iliac venous system, anterior to the sacroiliac joints. The left CIV is longer than the right one. The left CIV is located first medially, then posteriorly to the left EIA, whereas the right CIV is posterior and then lateral to the right EIA. The iliolumbar, the ascending lumbar vein, and the lateral sacral vein drain into the CIV. In most cases, the median sacral vein drains into the left CIV [41,42,47].

### 12.2. CIV Variations Related to the PLNDGO 

Surgical considerations. All types of CIV variations are related to possible venous injuries (Figure 13) [35,69,70,71,72,73,74]. Iatrogenic damage could occur during dissection of all five types of common iliac lymph node groups. Kose et al. reported a study of 229 patients who underwent paraaortic and PLND. Authors observed major retroperitoneal vessel variations in thirty nine (17%) patients. Great vessel injury was present in nineteen (8.3%) patients. CIV variations were found in two patients. One of the patients had a venous annulus of the right CIV, surrounding the right CIA. The other patient had a duplicated left CIV (B2 from the CIV classification), which was injured during dissection. Authors concluded that each patient’s vascular anatomy must be assessed individually to avoid injuries during scheduled operations [75].

### 12.3. CIV Tributaries Variations Related to PLNDGO

Iliolumbar vein (ILV) and ascending lumbar vein (ALV) anatomy. The ILV drains the venous blood from the iliac fossa, the iliac, and psoas muscles and terminates in the CIV. It is considered the segmental equivalent of the fifth lumbar vertebra [76,77,78,79,80,81]. The ALV participates in an anastomotic venous system between the inferior vena cava and the superior vena cava. The lower end of the ALV enters the cephalic border of the CIV. Upwards, the ALV receives the lumbar veins and terminates by joining the subcostal vein to form the azygos vein on the right and the hemiazygos on the left. ILV anastomoses with ALV, deep circumflex iliac vein and lateral sacral vein [76,77,78,79,80,81].

ILV and ALV variations related to PLNDGO. Particular attention should be given to the ILV and AVL as high percentage of drainage variations is documented. Furthermore, our literature survey revealed that the clinical significance of these veins is rarely mentioned in gynecologic oncology practice. In PLNDGO, we observed a high percentage of drainage variations of the ILV and ALV. In medical literature, there is a controversy as to the anatomy and the nomenclature of the ILV and ALV. Terms such as “lateral lumbosacral veins”, “inferior lumbar”, and “superior iliac” veins have been used to define ILV and ALV [76,77,78,79,80,81]. Moreover, different drainage patterns of ILV and ALV have been reported for both veins—ILV/ALV draining separately into CIV; ILV/ALV draining into the CIV as a common trunk, ILV draining into the external/internal iliac venous system. Lolis et al. reported a detailed description of the surgical anatomy and draining patterns of the ILV, based on a significantly great number of specimens [77]. They proposed and illustrated a detail classification separated into two types. In Type I (54%), ILV drainage patterns differed, whereas the ALV had the same pattern on both sides. In Type II, the ALV differed in pattern from one side to the other (46%). Authors observed high percentage of drainage variations in ILV 91% compare to ALV 34% [77]. Numerously drainage variations of ILV and ALV have been reported, but in Figure 14 are illustrated the most important during PLNDGO. The ILV and ALV drainage variations during PLND in our practice are shown in Figure 15.

Surgical considerations. Knowledge of the surgical anatomy of ILV and ALV may prevent venous damage such as tears and avulsion of these veins during PLND. Injury of ILV and ALV could occur in the course of external and internal iliac lymph nodes dissection. Special attention should be paid during middle common iliac (located in the lumbosacral fossa) and lateral external iliac lymph nodes dissection. Panici et al. stated that during lateral common iliac lymph nodes dissection, the presence of iliolumbal veins could be hazardous as several iliolumbar veins could drain into the CIV. Authors concluded that the CIV should be handled very gently, and dissection must be blunt and delicate [17].

## 13. Median Sacral Vein (MSV) and Lateral Sacral Veins (LSVs) Anatomy and Variations Related to PLNDGO

The median sacral vein (MSV) runs anterior and in the midline of the sacrum and the coccyx. It commonly drains into the left CIV. The lateral sacral veins lie on the periosteum of the sacrum and typically connect the epidural plexus with the internal iliac veins. The MSV might drain into the left internal iliac vein or the common iliac junction. Anastomoses between the lateral and median sacral veins form the presacral venous plexus. Cardinot et al. reported a case of both internal iliac veins, which formed a common trunk with a short and an average course, receiving the middle sacral vein’s drainage and flowing into the left external iliac vein. The lateral sacral veins (LSVs) might drain into the CIV and external iliac vein [41,47,72,82].

Surgical considerations. The MSV and the LSVs have to be preserved in cases of presacral and lateral sacral lymph node group dissection. Surgeons should be aware of different drainage patterns and venous plexus existence between the two veins.

External iliac vein (EIV) anatomy. The EIV is the continuation of the femoral vein. The inferior epigastric vein, the deep circumflex iliac vein, and the pubic branch drain into the EIV. The vein is located medially to the ipsilateral homonymous artery.

## 14. EIV Variations Related to PLNDGO

Anatomical variations of the EIV are less common than the CIV and internal iliac vein. The EIV might double, be absent, or be located lateral to the homonymous artery [82,83,84]. Hayashi et al. reported a case of an additional right EIV, which originated 45 mm inferior to the iliocaval junction and ran ventrally to the EIA to surround it with a right EIV. The right CIV was absent [84]. Djedovic and Putz observed a case of a venous annulus of the left external iliac vein. The medial and the lateral branch of the left EIV surrounded the left EIA. Moreover, a communication branch, between the lateral and the medial branches of the EIV, was identified. It was located below the left EIA [83].

Surgical considerations. Lateral, additional, double EIV, or venous annulus might be injured during a dissection of external iliac lymph nodes. Lateral, middle, and median external iliac lymph nodes are at great risk of iatrogenic damage. EIV injuries during PLNDGO have been reported in medical literature [85,86]. Roda et al. reported two cases of EIV injury among 327 pelvic lymphadenectomies for gynecological malignancies [86]. Kose et al. reported a case where damage to the EIA was due to supernumerary renal artery and vein, which distorted the normal anatomy [75].

## 15. EIV Tributaries Variations Related to PLNDOG

Deep circumflex iliac vein (DCIV) anatomy and variations related to PLNDG. The deep circumflex iliac vein (DCIV) runs over the EIA and above the inguinal ligament, it drains in the EIV. It is known that the draining pattern of the DCIV is variable. Ghassemi et al. observed that the DCIV ran over (82.5%) and under (17.5%) the EIA. Their study included 216 hemipelvises—78 cadavers and 60 clinical cases [87].

Surgical considerations. The DCIV under the EIA is less likely to be identified. Such an instance may lead to an expanded pelvic lymph node dissection (PLND). As mentioned above, the DCIV is the ventral border of PLND. The DCIV under the EIA is visualized between the EIA and EIV or by medial traction of the EIV (Figure 16).

## 16. Internal Iliac Vein (IIV) Anatomy

IIV follows its named arterial counterpart and ascends posteromedial to the IIA. IIV drains towards the ipsilateral EIV. The IIV tributaries are the superior/inferior gluteal, obturator, internal pudendal, lateral sacral, middle rectal, superior/inferior vesical, uterine, and vaginal veins. The retroperitoneal venous system is derived from the modification of three parallel primary venous networks in the embryo between the sixth and tenth weeks of gestation—the subcardinal, the postcardinal, and the supracardinal veins [41,42,88,89].

## 17. IIV Variations Related to PLNDGO

The multiple anomalies in the hypogastric venous drainage system represent posterior cardinal vein maldevelopment, as the posterior cardinal veins form the iliac bifurcation and iliac veins [75]. IIV variations have not been studied as thoroughly as IIA variations [79,89,90,91,92]. Despite the various classifications describing the diverse variations of the IIV, there is no established standard classification [79,89,90,91,92,93,94]. Shin et al. have created an impressive, comprehensive, and generally reliable classification of iliac vessel variations based on 2488 patients using multidetector computed tomography [90]. However, other types of IIV variations exist, which have not been mentioned in Shin’s classification. Therefore, to clarify most of IIV variations, a modification of Shin’s classification was developed based on previous findings (Figure 17 and Figure 18) [42,73,79,89,90,91,92,93,94].

Surgical considerations. As shown in Figure 17 and Figure 18, the prevalence of IIV variations is high. These variation veins may cause problems of unexpected hemorrhage during dissection of all lymph node groups. Therefore, it is crucial to recognize the presence of these variations, which are often encountered intraoperatively. Gyimadu et al. reported three (8.1%) cases of left duplicated IIV injuries (E2 from the IIV classification) among 37 patients with anatomical vessel variations. All of the patients underwent PLND and paraaortic lymphadenectomy for gynecological malignancies. Authors concluded that anatomical vessel variations are not uncommon and may increase the risk of vascular complications during PLND [67]. Panici et al. described the frequency of retroperitoneal variations among 309 consecutive patients with cervical, endometrial, and ovarian cancer treated with systematic aortic and PLND. Authors observed three (1.3%) cases of right IIV draining into the left CIV (B2 from the IIV classification). There were no cases of intraoperative injury to these veins [37].

Anatomical variations of iliac veins—conclusion of surgical considerations.

Iliac vein variations and injuries are more common than arterial ones. The three-dimensional (3D) models are reconstructed on the basis of multi-detector computed tomography. The surgeons observe the reconstructed 3D models and identify all of the anatomical structures before surgery. The 3D models of the pelvic vessels may help avoid injury to anatomical vessel variations during PLNDGO by providing information on individual anatomical features before gynecological procedures [95].

## 18. Corona Mortis, Aberrant and Accessory Obturator Veins Related to PLNDGO

Corona mortis (CMOR) is defined as any vessel anastomoses between the external iliac and obturator vessels, excluding aberrant and accessory obturator vessels. These originate from the external iliac or the inferior epigastric vessels and pierce the obturator membrane, not participating in anastomoses [55,96]. The CMOR could be arterial, venous anastomoses, or both. The frequency of venous CMOR is higher than the arterial. The CMOR is located behind the superior pubic ramus and on the posterior aspect of the lacunar ligament [55,96,97,98].

Aberrant and accessory obturator veins could arise from the EIV and its tributaries. The definition of an aberrant obturator vein is a vein that drains into the EIV system. There is no other obturator vein (Figure 19).

An accessory obturator vein is an extra obturator vein, draining into the EIV system, in addition to the normal counterpart [47,96,97,98].

Surgical consideration. Damaging the CMOR, aberrant and accessory obturator vessels could occur throughout medial external iliac and obturator lymph node group dissection. Aberrant or accessory obturator veins have vertical direction through the obturator canal. Injuring of these vessels is more troublesome than CMOR injury, as obturator nerve and artery are located nearby and should be preserved. Lee et al. reported two (10.5%) cases of aberrant obturator veins injury during 19 PLND for gynecological malignancies [99]. Selcuk et al. reported four (4.1%) cases of CMOR injuries among 209 patients who underwent PLNDGO [100].

## 19. Nerves Anatomy

### 19.1. Obturator Nerve (ON) Anatomy

The obturator nerve (ON) arises from the ventral roots of the second, third, and fourth lumbar nerves. It descends through the fibers of the psoas major muscle and emerges from its medial border. The ON crosses the sacroiliac joint behind the CIA, lateral to the internal iliac vessels travels along the lateral wall of the lesser pelvis and enters the obturator foramen. The ON is located cranial to the OA and OV [41,101,102,103,104].

### 19.2. ON Variations Related to PLNDGO

An accessory obturator nerve (AON) could arise from the anterior divisions of L2–L3, L3 only, L3–L4, from the ON, and from the femoral nerve. The AON is located medially to the femoral nerve and laterally to the ON. The nerve lies on the medial border of the psoas major muscle, but instead of piercing the obturator foramen, it passes over the superior pubic ramus. It runs behind pectineus and divides into three branches, which are also variable. The incidence of AON in the human population varies from 10% to 30%. Studies did not find differences of AON presence between genders [41,101,102,103,104].

Surgical considerations. Compression and subsequent neuropathy may occur as a result of damage to the AON [103]. Such an injury is possible during a dissection of the lateral external iliac, obturator, lateral, and middle common iliac lymph nodes.

### 19.3. Genitofemoral Nerve (GFN) Anatomy

The origin of the genitofemoral nerve (GFN) is from the ventral rami of L1 and L2 of lumbar plexus. It penetrates the psoas major muscle and runs cranially along the anterior aspect of the muscle, beneath the transversalis fascia and the peritoneum. In most cases, the GFN bifurcates into its both branches midway along the anterior surface of the psoas major. The genital branch follows the inguinal ligament and ends in the skin of mons pubis and labium majus. The femoral branch leaves the pelvis by passing through the femoral sheath lateral to the femoral artery and supplies the skin of the proximal anterior thigh [41,102,105,106,107].

### 19.4. The GFN Variations Related to PLNDGO

The GFN exhibits a large number of origin variations—T12-L1, L2-L3, L1, L2, and L3. Unilateral absence of the GFN has been reported. In such cases, the ilioinguinal nerve replaces the genital branch and the anterior femoral nerve or lateral cutaneous replaces the femoral branch. The genital or femoral branches of the nerve may arise separately [101,105,106,107]. Paul and Shastri observed the GFN in 60 hemipelvises. They reported for early division of the nerve into genital and femoral branches at its formation in 13.3% of hemipelvises or in the middle of its course, after emerging from psoas major in 3.3% of specimens [106]. Another study, reported that the most common variation of the GFN was splitting of the nerve into genital and femoral branches within the substance of the psoas muscle [102]. Injury to the GFN may cause entrapment neuropathy [106].

Surgical considerations. As the GFN is the lateral border of PLND, it should be identified on the psoas major muscle prior to PLND. Early division of the GFN into genital and femoral branches means that two nerve fibers would be identified on the psoas major muscle—genital and femoral. If the two nerve fibers are recognized on the psoas major muscle, they should be preserved to prevent neuropathy.

Anatomical variations of the GFN and the ON—conclusion of surgical considerations.

Cardosi reported a study of 1210 patients, who underwent major pelvic surgeries for gynecological malignancies. Twenty-three patients had postoperative neuropathies. The incidence of obturator nerve injury (39% of all neuropathies) was higher than for other nerve lesions. Genitofemoral neuropathy was identified in four (17.3% of all neuropathies) women who underwent PLND. The frequency of injury of variant ON and GFN during PLNDGO is uncertain, but it is believed to be higher than those with normal anatomy [108].

There are several strengths of the present article. First, such a comprehensive review of the topic has never been made. Second, despite the multitude of articles describing PLNDGO, authors did not mention differences in morphology of the EIA. Very few anatomical articles reported morphological differences of the EIA [48,53,54]. Third, the different drainage patterns of ILV and ALV have never been discussed in gynecologic oncology. An article presented by Cibula and Rustum illustrated the ILV draining into the EIV and the CIV [5]. Panici et al. discussed the importance of ILV draining into the CIV during PLNDGO [17]. In both articles, it is not mentioned that the ALV could also drain into the CIV, EIV, or IIV. These articles described the ILV as the only vein draining into the EIV or the CIV. Moreover, the ILV and the ALV may drain into the iliac venous system by sharing a common trunk.

A potential limitation of the present article is that some of the anatomical variations are rare and there is limited data about the actual incidence of complications during PLNDGO. A possible explanation about the limited data could be that injuries to variant anatomical structures are managed during surgery. Furthermore, injuries with fatal outcome are less likely to be reported. We encourage surgeons to share their experience with injuries to variant anatomical structures during PLNDGO in order to estimate the actual incidence of complications.

## 20. Conclusions

A wide variety of anatomical variations among pelvic structures (ureters, vessels, and nerves) could cause severe and potentially lethal complications during surgery. The majority of the anatomical variations are discovered intraoperatively. Therefore, a detailed knowledge of the anatomy and anatomical variations is essential in order to prevent serious damage to vital structures during pelvic operations. The present article aims to expand the limited knowledge about anatomical variations in the pelvis. An association between variations of the most important pelvic structures and PLND is conducted for the first time. We hope that the detailed review of the anatomical variations will decrease patient morbidity and mortality. Furthermore, accurate description and analysis of the majority of pelvic anatomical variations may impact not only gynecological surgery, but also spinal surgery, urology, and orthopedics.

## Figures and Tables

**Figure 1 diagnostics-11-00089-f001:**
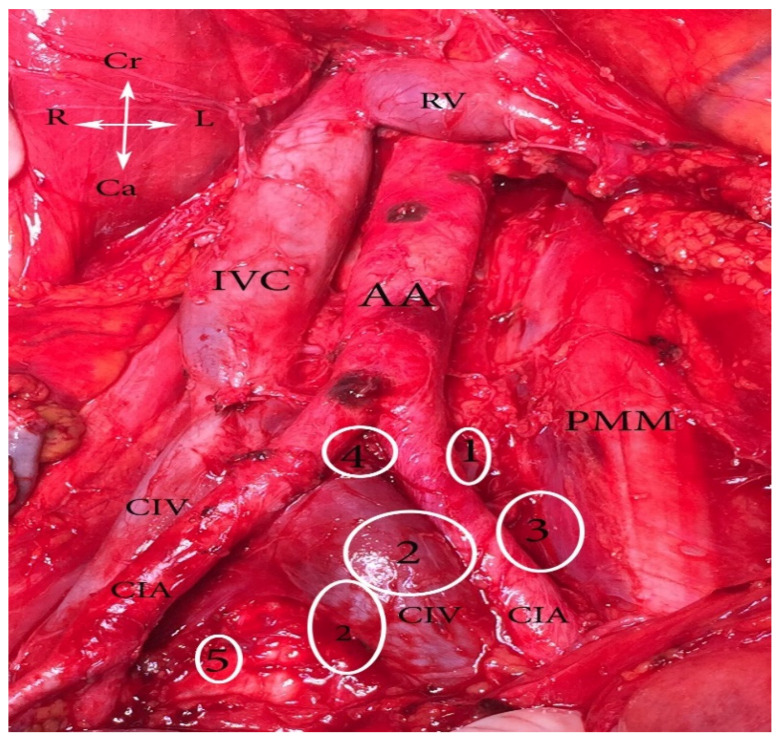
Common iliac lymph nodes classification (open surgery). **1.** Lateral—between lateral part of CIV and medial part of psoas major muscle, **2.** medial—medial to CIV and CIA, **3.** middle—located in the lumbosacral fossa, **4.** subaortic—below aortic bifurcation, **5.** promontory—at the promontory. AA—abdominal aorta, IVC—inferior vena cava, RV—right renal vein, PMM—psoas major muscle, CIA—common iliac artery, CIV—common iliac vein, Cr—cranial, Ca—caudal, L—Left, R—right.

**Figure 2 diagnostics-11-00089-f002:**
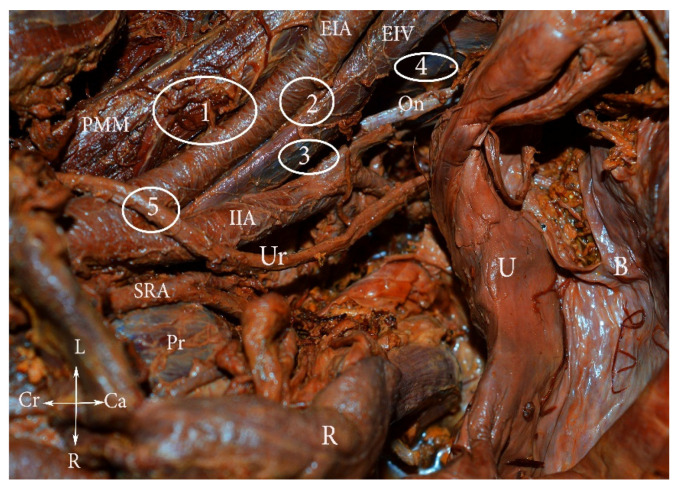
External iliac lymph nodes classification (embalmed cadaver). **1.** Lateral—lateral to external iliac artery, **2.** middle—medial to the EIA and lateral to the EIV, **3.** medial—medial to both external iliac vessels, **4.** obturator—around the obturator nerve and vessels, **5.** interiliac—at the level of CIA bifurcation, between the EIA and IIA. PMM—psoas major muscle, EIA—external iliac artery, EIV—external iliac vein, IIA—internal iliac artery, Ur—ureter, U—uterus, B—bladder, SRA—superior rectal artery, Pr—promontorium, R—rectum, L—left, r—right, Cr—cranial, Ca—caudal.

**Figure 3 diagnostics-11-00089-f003:**
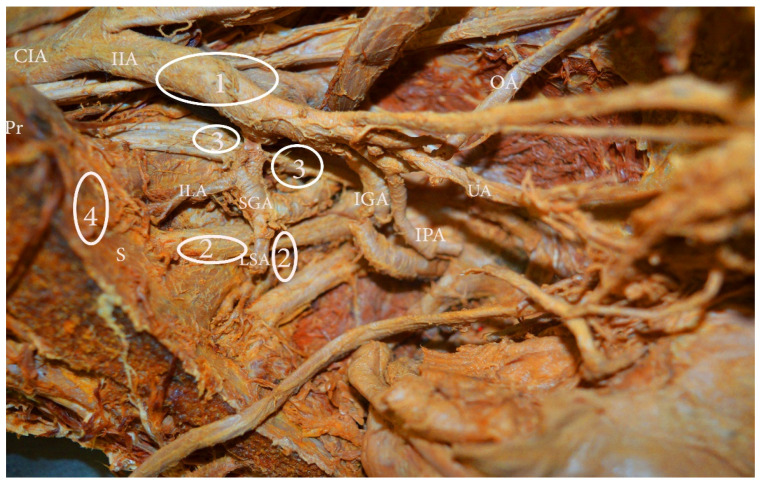
Internal iliac lymph nodes classification (embalmed cadaver—left hemipelvis). **1.** Anterior—anterior to anterior division of internal iliac artery, **2.** lateral sacral—close to the paired lateral sacral arteries, **3.** gluteal—between superior gluteal and internal iliac artery, **4.** sacral (presacral)—along median sacral artery. CIA—common iliac artery, IIA—internal iliac artery, OA—obturator artery, UA—umbilical artery, IPA—internal pudendal artery, IGA—inferior gluteal artery, LSA—lateral sacral artery, ILA—iliolumbar artery, Pr—promontorium, S—sacrum.

**Figure 4 diagnostics-11-00089-f004:**
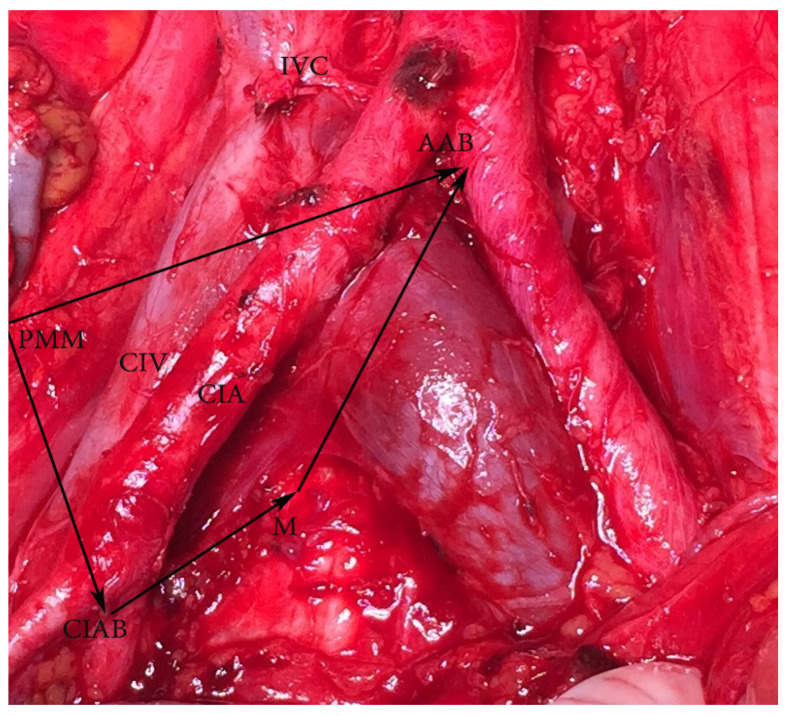
Common iliac lymph nodes dissection—anatomic boundaries (open surgery, right side). Dorsal—abdominal aorta bifurcation, ventral—common iliac artery bifurcation, medial—medial aspect of common iliac vessels (on the left side is mesoureter), lateral—psoas major muscle. AAB—abdominal aorta bifurcation, PMM—psoas major muscle, CIAB—common iliac artery bifurcation, M—medial aspect of common iliac vessels on the right side (on the left side is mesoureter), IVC—inferior vena cava, CIA—common iliac artery, CIV—common iliac vein.

**Figure 5 diagnostics-11-00089-f005:**
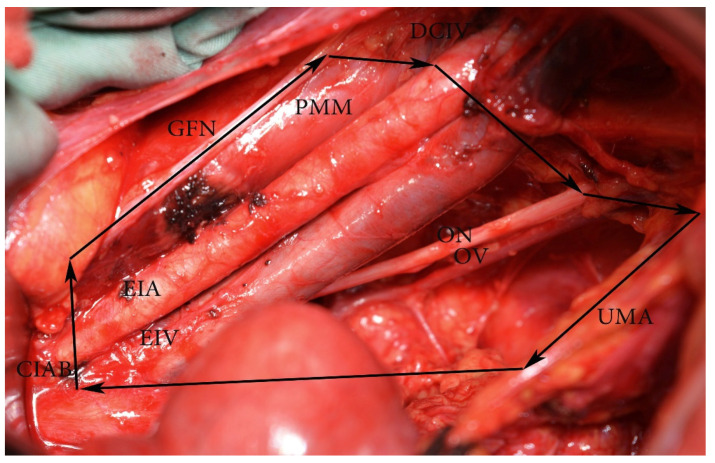
External and internal iliac lymph nodes dissection—anatomic boundaries (open surgery left pelvic sidewall). CIAB—common iliac artery bifurcation, ON—obturator nerve, DCIV—deep circumflex iliac vein, GFN—genitofemoral nerve, PMM—psoas major muscle, EIA—external iliac artery, EIV—external iliac vein, OV—obturator vein. The medial border is the ureter dorsally and obliterated umbilical artery ventrally. In the figure, the ureter is stretched medially for better exposure of the visible structures.

**Figure 7 diagnostics-11-00089-f007:**
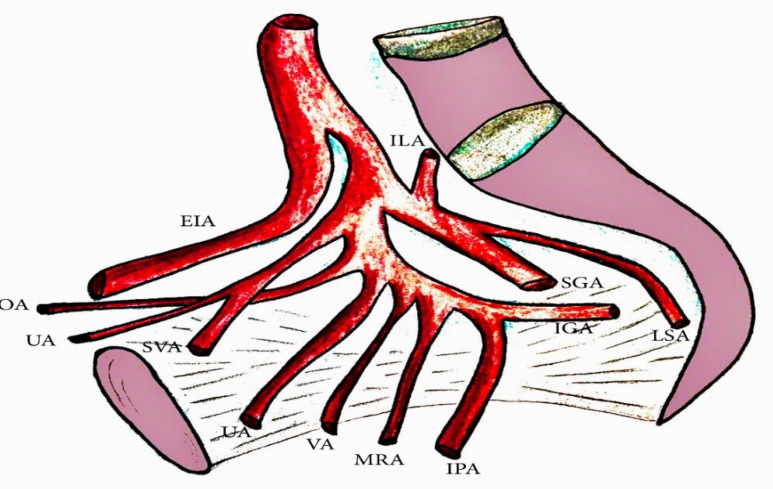
The internal iliac artery (IIA) branches. OA—obturator artery, UA—obliterated umbilical artery, SVA—superior vesical artery, UA—uterine artery, VA—vaginal artery, MRA—middle rectal artery, IPA—internal pudendal artery, IGA—inferior gluteal artery, LSA—lateral sacral artery, SGA—superior gluteal artery, ILA—iliolumbar artery.

**Figure 8 diagnostics-11-00089-f008:**
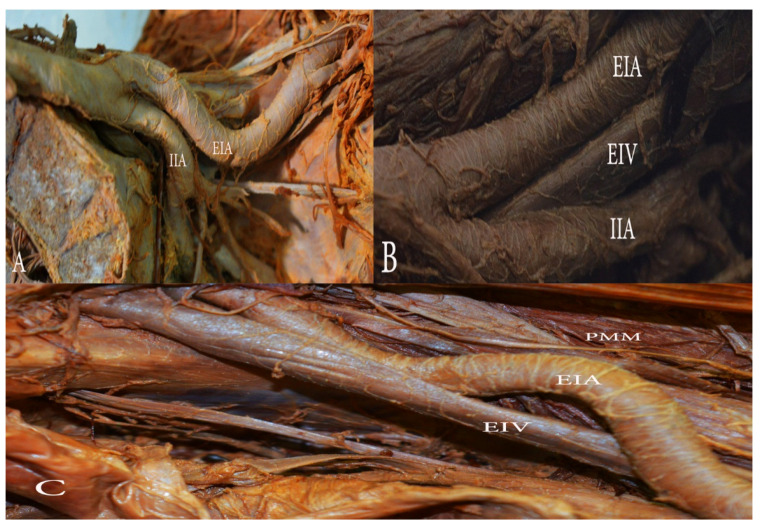
Morphological variations of the external iliac artery (EIA). (**A**) The EIA with an inward loop in the left hemipelvis. (**B**) The EIA with a gentle inward loop in the left hemipelvis. (C) ’S’ shaped EIA in the right hemipelvis. EIA—external iliac artery, EIV—external iliac vein, IIA—internal iliac artery, PMM—psoas major muscle.

**Figure 9 diagnostics-11-00089-f009:**
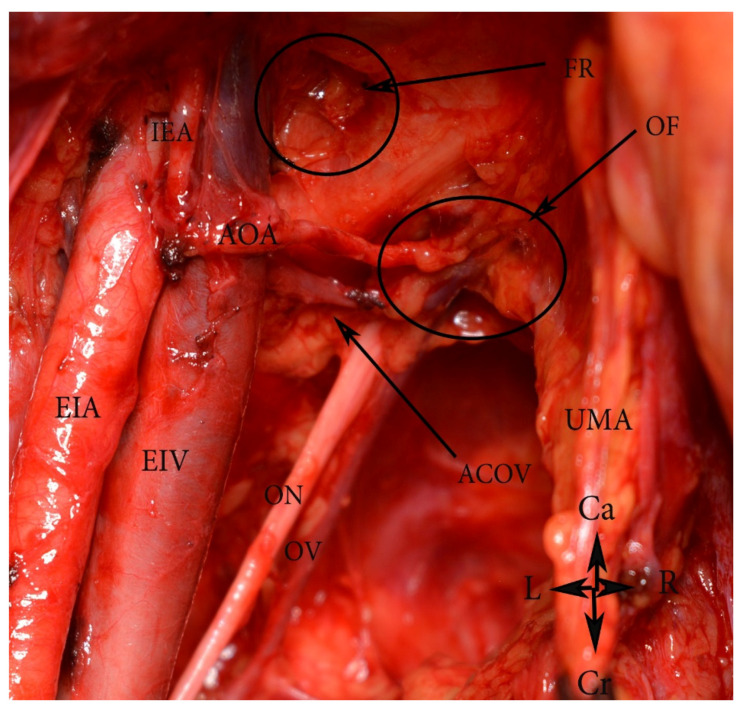
An aberrant obturator artery arising as a common trunk with the inferior epigastric artery (open surgery). EIA—external iliac artery, EIV—external iliac vein, AOA—aberrant obturator artery, ACOV—accessory obturator vein, ON—obturator nerve, OV—obturator vein, IEA—inferior epigastric artery, UMA—umbilical artery, FR—the deep femoral ring, OF—obturator foramen. Ca—caudal, Cr—cranial, R—right, L—left.

**Figure 10 diagnostics-11-00089-f010:**
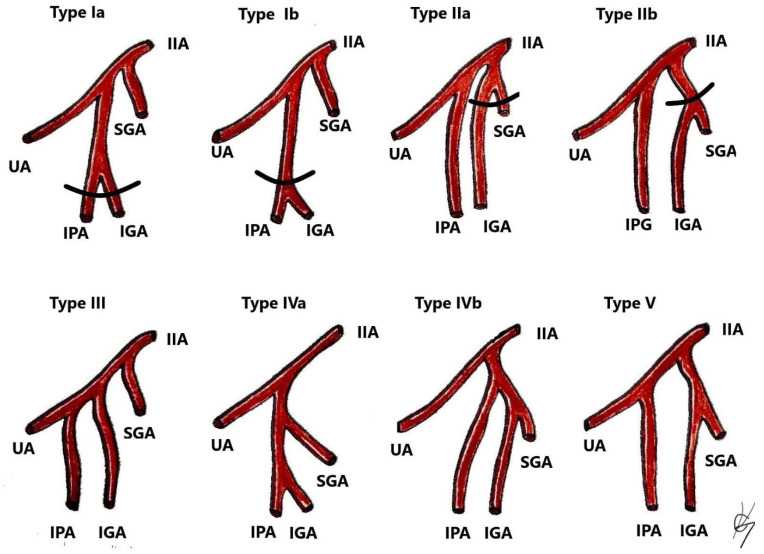
Classification of IIA variations. Adopted from Adachi [58]. Type I—The superior gluteal artery (SGA) arises separately from internal iliac artery, while the inferior gluteal (IGA) and internal pudendal vessel (IPA) share a common trunk. Type Ia—the bifurcation of IGA and IPA occurs within the pelvis. Type Ib—the bifurcation occurs below the pelvis. Type II—The internal pudendal artery arises separately from the IIA, while the superior gluteal artery shares a trunk with the inferior gluteal artery. Type IIa—the bifurcation of SGA and IGA occurs within the pelvis. Type IIb—the bifurcation occurs below the pelvis. Type III—SGA, IGA, and IPA arise separately from the internal iliac artery, and the internal pudendal artery is the last branch. Type IV—SGA, IGA, and IPA share a common trunk. Type IVA—the SGA is the first vessel from the common trunk, before bifurcating into the other two branches—SGA and IGA. Type IVB—the IPA is the first from the common trunk, which then divides into SGA and IGA. Type V—The IGA has a separate origin from the IIA, while the SGA and IGA share a common trunk.

**Figure 11 diagnostics-11-00089-f011:**
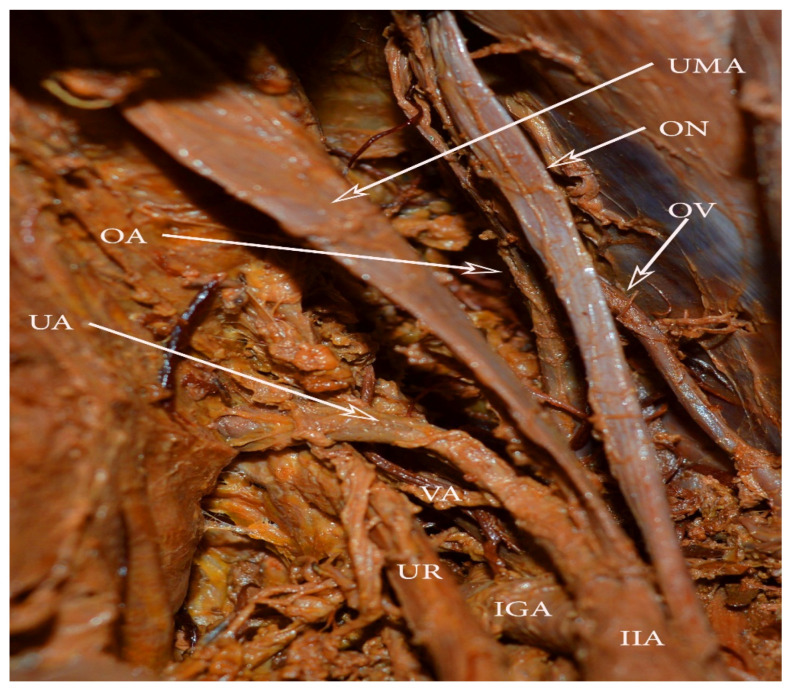
Uterine artery and umbilical artery, arising in a common trunk. VA arising from the uterine artery. OA—obturator artery, OV—obturator vein, ON—obturator nerve, UMA—obliterated umbilical artery, IIA—internal iliac artery, IGA—inferior gluteal artery, UR—ureter, VA—vaginal artery.

**Figure 12 diagnostics-11-00089-f012:**
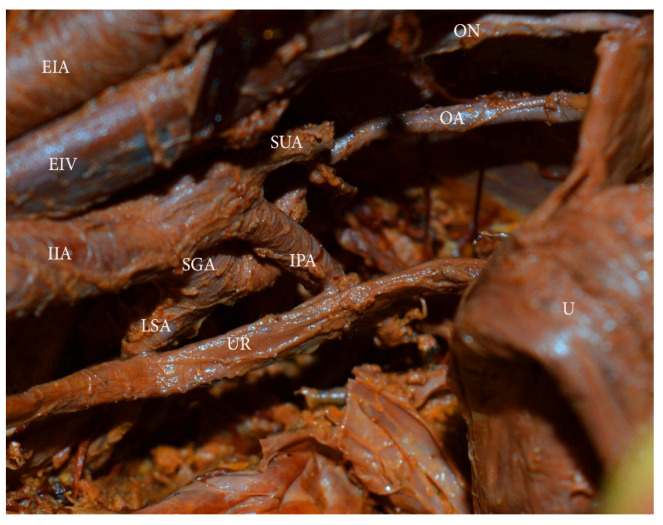
Obturator artery arising from the posterior branch of the IIA. U—uterus, EIA—external iliac artery, EIV—external iliac vein, IIA—internal iliac artery, SGA—superior gluteal artery, LSA—lateral sacral artery, IPA—internal pudendal artery, OA—obturator artery, ON—obturator nerve, SUA—severed uterine artery.

**Figure 13 diagnostics-11-00089-f013:**
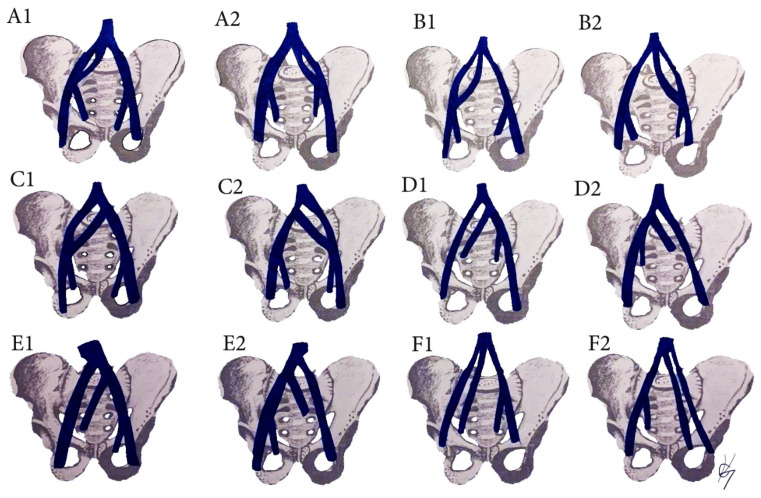
Common iliac vein variations. (**A**) Incomplete duplication of the CIV; (**B**) complete duplication of the CIV; (**C**) lateral duplicated branch drains into the IVC, the medial drain into the CIV. (**D**) Absent CIV, external and internal iliac veins drain to the contralateral CIV; (**E**) absent CIV, the EIV drains into the IVC, the IIV drains into the contralateral CIV; (**F**) Absent CIV, the external and internal veins drain into IVC. Inferior vena cava (IVC), Common iliac vein (CIV), external iliac vein (EIV), internal iliac vein (IIV). (**A1**,**B1**,**C1**,**D1**,**E1**,**F1**) are related to right hemipelvises variations, whereas (**A2**,**B2**,**C2**,**D2**,**E2**,**F2**) are related to left hemipelvises variations.

**Figure 14 diagnostics-11-00089-f014:**
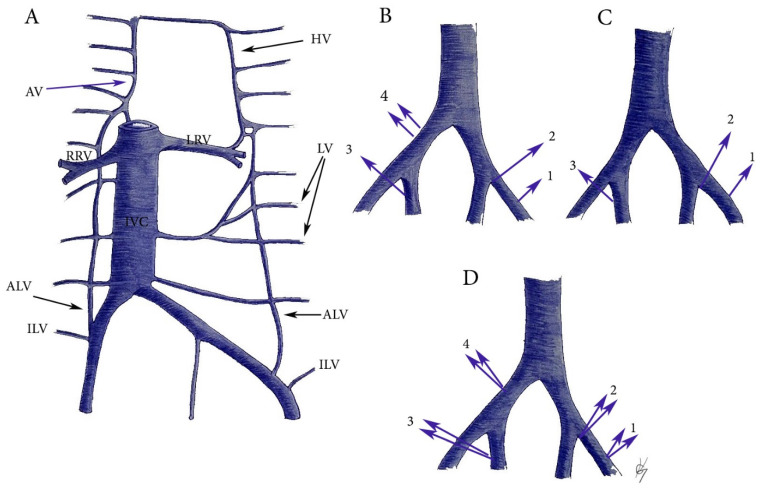
ILV and ALV anatomy and variations. (**A**) ILV and ALV anatomy. HV—hemiazygos vein, LV—lumbar veins, ALV—ascending lumbar vein, ILV—iliolumbar vein, IVC—inferior vena cava, LRV—left renal vein, RRV—right renal vein, AV—azygos vein. (**B**) ILV variations. 1—drains into EIV, 2—drains into the confluence of the CIV, 3—drains into the IIV, 4—two ILVs drains into the CIV. (**C**) ALV variations. 1—drains into the EIV, 2—drains into the confluence of the CIV, 3—drains into the IIV. (**D**) Common trunks between ALV and ILV. 1—drains into the EIV, 2—drains into the confluence of CIV, 3—drains into the IIV, 4—drains into the CIV.

**Figure 15 diagnostics-11-00089-f015:**
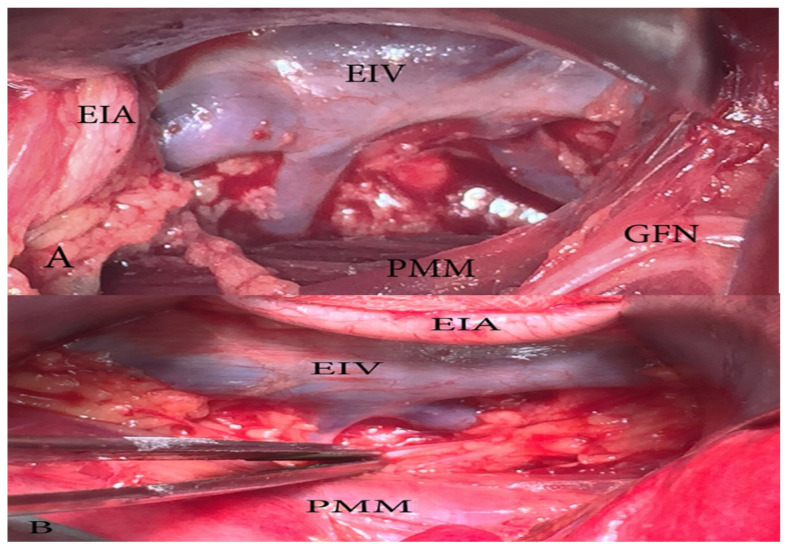
ILV or ALV drain into the EIV (open surgery right pelvic sidewall). We can only speculate if these veins are ILV, ALV, or both. (**A**) Two separate veins drain into the EIV. The EIA is retracted medially. (**B**) Two veins drain into the EIV via common trunk.

**Figure 16 diagnostics-11-00089-f016:**
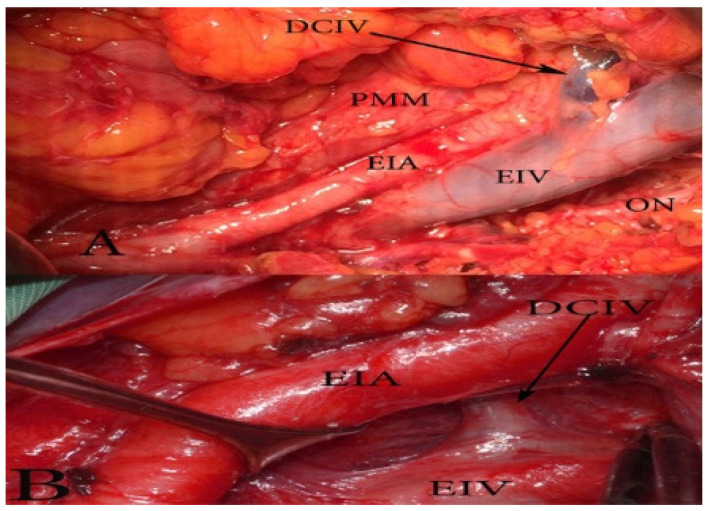
The deep circumflex iliac vein (DCIV) normal anatomy (**A**) and variation (**B**) (open surgery, left pelvic sidewall). (**A**) The DCIV runs over the EIA and drains into the EIV. (**B**) The DCIV passes under the EIA and drains into the EIV.

**Figure 17 diagnostics-11-00089-f017:**
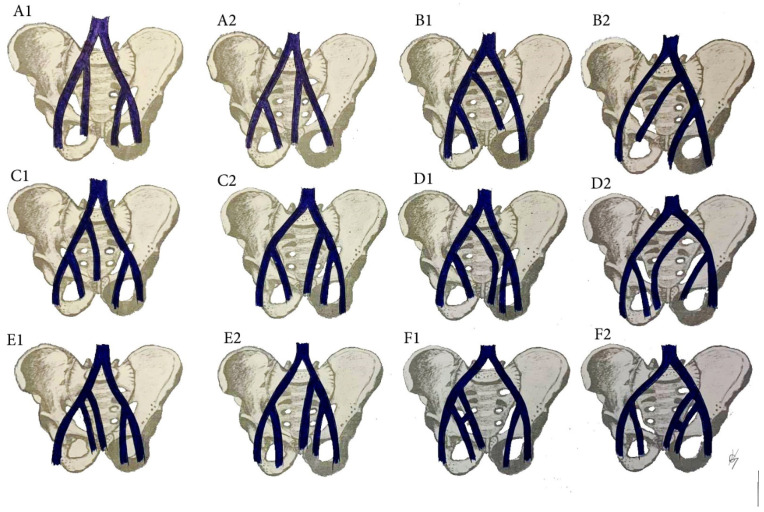
IIV variations. (**A1**) high joining of the IIV to the ipsilateral EIV. (**B1**) Joining of the IIV to the contralateral CIV. (**C1**) Separated trunk of the IIV drains into the ipsilateral CIV. (**D1**) separated trunk of the IIV drains into the contralateral CIV. (**E1**) Duplication of the IIV. (**F1**) Duplication of the IIV with a venous connection between them. Variations 1 are related to right pelvic sidewall, whereas variations 2 are related to the left pelvic sidewall.

**Figure 18 diagnostics-11-00089-f018:**
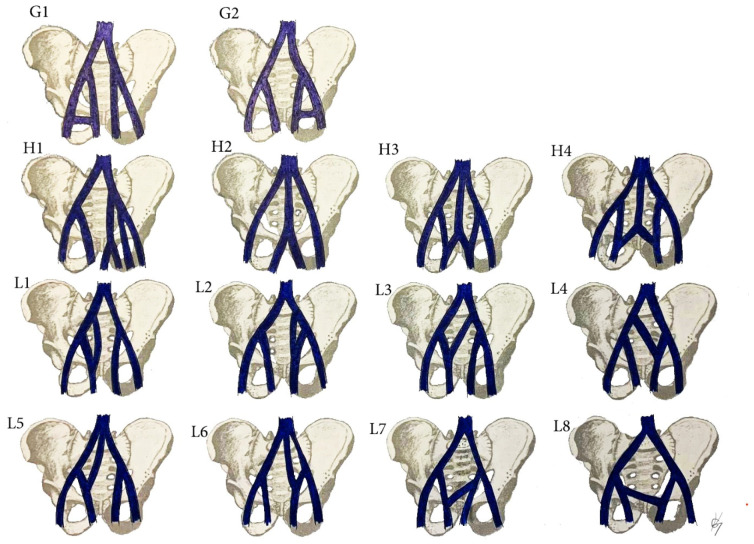
IIA variations. (**G1**) Communication vein between the IIV and the EIV. (**H1**) Separated trunk of bilateral internal iliac veins connecting with each other before draining into the left CIV. (**H2**) The internal iliac veins form a common trunk, that drains into the inferior vena cava. (**H3**) The internal iliac veins form a common trunk, which drains into the inferior vena cava, communication vein between the IIVs and ipsilateral EIV. (**H4**) Both IIVs are joined by a connecting vein that drains into the IVC. (**L**) Communication veins. (**L1**,**L2**) Communication vein between the IIV and ipsilateral CIV. (**L3**,**L4**) communication vein between the IIV and contralateral CIV. (**L5**,**L6**) Both internal iliac veins are joined with a communication vein, which drains into the inferior vena cava. (**L7**,**L8**) Communicating vein between both IIVs. Variations (**G1**,**L1**,**L3**,**L5**,**L7**) are related to right pelvic sidewall, whereas variations (**G2**,**L2**,**L4**,**L6**,**L8**) are related to left pelvic sidewall.

**Figure 19 diagnostics-11-00089-f019:**
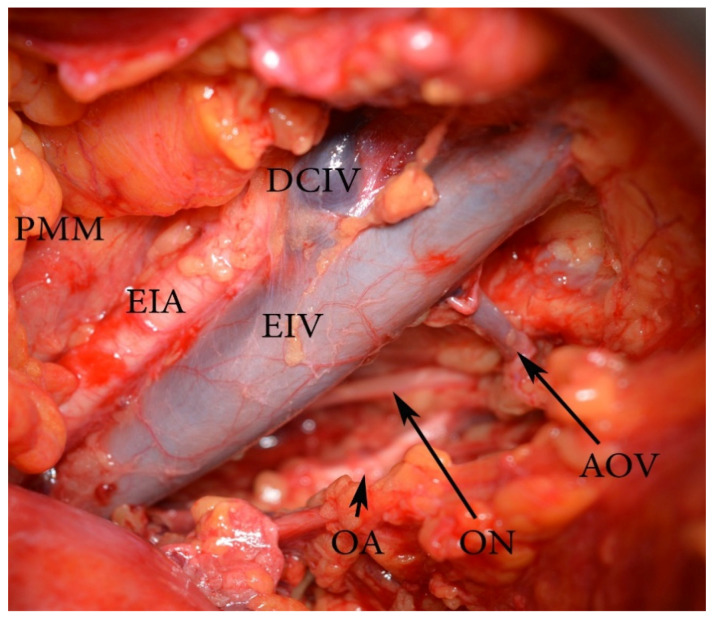
An aberrant obturator vein (left pelvic sidewall). EIA—external iliac artery, EIV—external iliac vein, PMM—psoas major muscle, AOV—aberrant obturator vein, ON—obturator nerve, OA—obturator artery, DCIV—deep circumflex iliac vein.

## Data Availability

Authors declare that all related data are available concerning researchers by the corresponding author’s email.

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
