# Peer review of "Pelvic Lymphadenectomy in Gynecologic Oncology—Significance of Anatomical Variations"

_diagnostics, 2021, doi:10.3390/diagnostics11010089_

Round 1

Reviewer 1 Report

The research Manuscript diagnostics-1037133 entitled “Anatomical variations in the pelvis related to pelvic lymphadenectomy in gynecologic oncology" by Kostov et al. is a review article aimed to to expand the limited knowledge about anatomical variations in the pelvis; in this review the authors illustrate and explain the main variations of ureter, pelvic vessels and nerves and their association to pelvic lymphadenectomy.
Overall, the manuscript is well written, but in my opinion it does not add relevant novel insight to notions and other similar description already available in literature. Its major merit is to summarizing in a unique text the main anatomical variations and their surgical implications, thus giving to surgeons a further useful text for consultation for clinical practice.

Moreover, surgical considerations could be improved by accompanying the description of anatomical variants  with appropriate citations  the actual incidence of complications during lymphadenectomy and gynecologic oncology in relation to the specific anatomic variation described.
I recommend a careful revision of English.

Author Response

Answer to Reviewer 1.

  1. Overall, the manuscript is well written, but in my opinion, it does not add relevant novel insight to notions and other similar description already available in literature. Its major merit is to summarizing in a unique text the main anatomical variations and their surgical implications, thus giving to surgeons a further useful text for consultation for clinical practice.

Author’s Reply: We do not add relevant novel insight as it is a review article, not a research or original article. Although it is a review article, we think that there is a new and important information. For instance:

  1. Such an extend review about the topic has never been made before.
  2. EIA morphology variations have been described only in a few papers. We provided photos and discussed EIA morphology variations related to PLND in gynecologic oncology. We did not find such an article in literature. Only anatomical articles referred to it.
  3. Ascending lumbar vein and iliolumbar vein significance of anatomical variations have never been mentioned in gynecologic oncology, although there are reports only describing iliolumbar vein. These reports described the ILV as the only vein draining in the EIV. That is not accurate information. Only a few anatomical articles described variations patterns of these veins. There are no surgical articles defining the presence of these veins.

Our purpose was to explain, summarize and illustrate the inevitable relationship between ANATOMY and SURGERY. As onco-gynecologists, we believe that anatomy is the cornerstone in improving surgical skills. If surgeons knows anatomy and anatomical variations, he fells more confident in the operating room. We hope that our article will rise debate and provoke other surgeons to discuss the topic in the future.

  1. Moreover, surgical considerations could be improved by accompanying the description of anatomical variants with appropriate citations the actual incidence of complications during lymphadenectomy and gynecologic oncology in relation to the specific anatomic variation described.

Author’s Reply:

We completely agree with the reviewer. Unfortunately, some of the anatomical variations are rare and there is limited data about the actual incidence of complications. Moreover, the incidence of vascular injuries during gynecologic operation is largely unknown, and very few case reports that exist probably underestimate the problem. One reason could be that the vascular problems are solved during surgery and injuries with fatal outcome might be more likely to be reported.  Even papers dealing with complications to laparoscopy did not always define the type of vascular injury, and a detailed analysis is therefore impossible. We included another section at the end of ureters, arteries, veins and nerves variations sections and named it ‘’ Conclusion of surgical considerations”. To improved surgical considerations, we added the actual incidence of complications related to normal anatomy. It is supposed that the percentage of injuries related to anatomical variations are high. We also found papers dealing with lymphadenectomy and anatomical variations. They did not described separately every complications. Instead, they summarized complications into – ureteral, arterial and venous.

 Additionally, we found case reports describing injuries of some of the anatomical variations. Moreover, we found in some of the variations the actual incidence of complications related to the specific anatomic variations and incorporated in the text.

Ureteral variations – conclusion of surgical considerations

The next text was inserted:

Ureteral injuries (UIs) occur approximately in 0.5-1% of all pelvic operations with gynecological operations accounting for 75% of these. UIs are more common during radical hysterectomy procedures combine with PLND. The incidence of UIs during PLND is 10 %. It is believed that ureteral variations lead to an increased risk of UIs. There is consensus that a multiplication of the ureter is an independent risk factor for UIs during PLND. Benedetti-Panici et al. reported a prospective study, which included 309 consecutive patients with cervical, endometrial, and ovarian cancer treated with systematic aortic and pelvic lymphadenectomy. Ureteral duplication was observed in four (1.6%) cases. Authors concluded that knowledge of ureteral anomalies is important in gynecological surgery, as the risk of UIs is elevated. Duplicated UIs during gynecological procedures are reported in medical literature. UIs may occur at the level of the infundibulopelvic ligament and deep in the pelvis, below the level of ischial spine, where ureter lies lateral to the peritoneum of uterosacral ligament. Anatomical variations of the ureter are often associated with other congenital anomalies. Patients with known congenital anomalies or different types of syndromes should undergo preoperative imaging for careful surgical planning [33-39]. Postoperatively, a routine cystoscopy should be performed to rule out UIs [26].

 The changes are: Text on page 7. Lines: 212 - 228

References – {33-39} were inserted.

Iliac arteries 

EIA surgical considerations 

The next text was inserted: 

Surgical considerations. During an external iliac lymph nodes dissection, surgeons should be familiar with the EIA position differences to prevent iliac vessel injury. Safi et al. reported a case of EIA laparoscopic injury due to an anomalous position of the artery. The EIA was elongated and located posterior and medial to the EIV. They repaired the artery using a two-needle single-knot technique and continued the lymphadenectomy. Authors concluded that during PLND, both external iliac vessels should be exposed and visible to the surgeon, as the anatomical variations of the EIA increased the risk of injury [52].

The changes are: Text on page 10. Lines: 297-304.

Reference 52 was insereted.

  Anatomical variations of iliac arteries – conclusion of surgical considerations. 

The next text was inserted: 

Arterial injuries during PLNDGO are less common than venous ones. There are very few studies describing arterial damage during PLNDGO. Bae et al. presented a retrospective review study. Authors observed four (1.3%) cases of major vascular injuries during 225 laparoscopic lymph node dissections. One of the injuries was to the EIA. Ricciardi et al. and Ishikawa et al. also reported damage to the EIA in a course of PLND. These studies showed that the EIA is more likely to be injured during PLND than the other iliac arteries. Gyimadu et al. concluded that the occurrence of vascular injuries during the course of PLNDGO may be due to variations in the anatomy of great retroperitoneal vessels. In medical literature the frequency of such abnormalities varies between 5.6 and 23.0 % [64-68].

The changes are: Text on page 14-15. Lines: 390-399.

References 64-68 were inserted.

Iliac veins 

Common iliac vein variations 

The next text was inserted: 

Surgical considerations. All types of CIV variations are related to possible venous injuries. Iatrogenic damage could occur during dissection of all five types of common iliac lymph node groups. Kose et al. reported a study of 229 patients who underwent paraaortic and PLND. Authors observed major retroperitoneal vessel variations in thirty nine (17%) patients. Great vessel injury was present in nineteen (8.3%) patients. CIV variations were found in two patients. One of the patients had a venous annulus of the right CIV, surrounding the right CIA. The other patient had a duplicated left CIV (B2 from the CIV classification), which was injured during dissection. Authors concluded that each patient’s vascular anatomy must be assessed individually to avoid injuries during scheduled operations [75].

The changes are: Text on page 16. Lines: 424 -432 

The next reference was inserted: 75 

 ILV and ALV surgical consideration

The next text was inserted: 

 Panici et al stated that during lateral common iliac lymph nodes dissection, the presence of iliolumbal veins could be hazardous as several iliolumbar veins could drain into the CIV. Authors concluded that the CIV should be handled very gently, and dissection must be blunt and delicate [17].

The changes are: Text on page 18. Lines: 475- 478.

EIV surgical considerations 

The Next text was inserted: 

EIV injuries during PLNDGO have been reported in medical literature [85, 86]. Roda et al reported two cases of EIV injury among 327 pelvic lymphadenectomies for gynecological malignancies [86]. Kose et al reported a case where damage to the EIA was due to supernumerary renal artery and vein, which distorted the normal anatomy [75].

The changes are: Text on page 19. Line: 505- 510.

References 85-86 was inserted

IIV surgical considerations

The Next text was inserted: 

 Gyimadu et al. reported three (8.1%) cases of left duplicated IIV injuries (E2 from the IIV classification) among 37 patient with anatomical vessel variations. All of the patients underwent PLND and paraaortic lymphadenectomy for gynecological malignancies. Authors concluded that anatomical vessel variations are not uncommon and may increase the risk of vascular complications during PLND [67]. Panici et al. described the frequency of retroperitoneal variations among 309 consecutive patients with cervical, endometrial and ovarian cancer treated with systematic aortic and PLND. Authors observed three (1.3%) cases of right IIV draining into the left CIV (B2 from the IIV classification). There were no cases of intraoperative injury to these veins [37].

Anatomical variations of iliac veins – conclusion of surgical considerations.

Iliac vein variations and injuries are more common than arterial ones. The three-dimensional models (3D) are reconstructed on the basis of multi-detector computed tomography. The surgeons observe the reconstructed 3D models and identify all of the anatomical structures before surgery. The 3D models of the pelvic vessels may help avoid injury to anatomical vessel variations during PLNDGO by providing information on individual anatomical features before gynecological procedures [95].

The changes are: Text on page 21-22. Lines:561- 577. 

Reference 95 was inserted.

 Corona mortis, aberrant and accessory obturator veins surgical considerations. 

The next text was inserted: 

Lee et. al reported two (10.5%) cases of aberrant obturator veins injury during 19 PLND for gynecological malignancies [99]. Selcuk et al. reported four (4.1%) cases of CMOR injuries among 209 patients who underwent 96 PLNDGO [100].

The changes are: Text on page 22. Lines: 598- 600. 

References 99-100 were inserted. 

Anatomical variations of the GFN and the ON – conclusion of surgical considerations.

Cardosi reported a study of 1210 patients, who underwent major pelvic surgeries for gynecological malignancies. Twenty-three patients had postoperative neuropathies. The incidence of obturator nerve injury (39% of all neuropathies) was higher than for other nerve lesions. Genitofemoral neuropathy was identified in four (17.3% of all neuropathies) women who underwent PLND. The frequency of injury of variant ON and GFN during PLNDGO is uncertain, but it is believed to be higher than these with normal anatomy[108].

The changes are: Text on page 23. Lines 643 – 650.

Reference 108 was inserted. 

  1. I recommend a careful revision of English.

Author’s Reply: A native English speaker with medical edication carefully revised the whole manuscript. After discussion with the native speaker and between authors, we decided to change the Title. We think the new Title is better than the previous one.

Thank you very much for reviewing the article. We greatly appreciate you taking the time to review it.

Thank you,

Sincerely

Stoyan Georgiev Kostov, MD

Clinic of Gynecology

MHAT “Saint Anna”   ,Street “Car Osvoboditel” 100

Varna 9000, Bulgaria

Tel: +3590896711178

Reviewer 2 Report

I was favorably surprised by this manuscript. I can only extend my sincere congratulations to the authors, who have written a beautiful paper, complete and detailed, on a subject that is not widely considered in literature, but much listened to in the operating room. In fact, the anatomical variations found during pelvic lymphadenectomy in gynecologic oncology are very much discussed among surgeons, but little analyzed in the literature. This review fills a gap. Congratulations to the authors, for the excellent work done, including reported images (amazing). I have no other comments to report as the work can be accepted immediately.

Author Response

Answer to Reviewer 2.

I was favorably surprised by this manuscript. I can only extend my sincere congratulations to the authors, who have written a beautiful paper, complete and detailed, on a subject that is not widely considered in literature, but much listened to in the operating room. In fact, the anatomical variations found during pelvic lymphadenectomy in gynecologic oncology are very much discussed among surgeons, but little analyzed in the literature. This review fills a gap. Congratulations to the authors, for the excellent work done, including reported images (amazing). I have no other comments to report as the work can be accepted immediately.

Author’s Reply: Thank you very much for reviewing and approving the article. We greatly appreciate you taking the time to review it!

Thank you,

Sincerely

Stoyan Georgiev Kostov, MD

Clinic of Gynecology

MHAT “Saint Anna”   ,Street “Car Osvoboditel” 100

Varna 9000, Bulgaria

Tel: +3590896711178

Round 2

Reviewer 1 Report

I appreciated that revisions made by the authors that in my opinion improved the quality of the manuscript. As regard the Point 1, the authors could add briefly the main novelty and advances (i.e. points 2 and 3 of their replies to comment 1) of their study in comparison to the available literature in the Discussion section.

Author Response

Reviewer 1. I appreciated that revisions made by the authors that in my opinion improved the quality of the manuscript. As regard the Point 1, the authors could add briefly the main novelty and advances (i.e. points 2 and 3 of their replies to comment 1) of their study in comparison to the available literature in the Discussion section

Author’s Reply: We agree with the reviewer that the revisions significantly improved the quality of the manuscript. We added the main novelties and advances at the end of the Discussion section. We described the strengths and limitations of our manuscript.

The next text was inserted.

There are several strengths of the present article. First, such a comprehensive review of the topic has never been made. Second, despite the multitude of articles describing PLNDGO, authors did not mention differences in morphology of the EIA. Very few anatomical articles reported morphological differences of the EIA [48, 53, 54]. Third, the different drainage patterns of ILV and ALV have never been discussed in gynecologic oncology. An article presented by Cibula and Rustum illustrated the ILV draining into the EIV and the CIV [5]. Panici et al. discussed the importance of ILV draining into the CIV during PLNDGO [17]. In both articles, it is not mentioned that the ALV could also drain into the CIV, EIV or IIV. These articles described the ILV as the only vein draining into the EIV or the CIV. Moreover, the ILV and the ALV may drain into the iliac venous system by sharing a common trunk.

A potential limitation of the present article is that some of the anatomical variations are rare and there is limited data about the actual incidence of complications during PLNDGO. A possible explanation about the limited data could be that injuries to variant anatomical structures are managed during surgery. Furthermore, injuries with fatal outcome are less likely to be reported. We encourage surgeons to share their experience with injuries to variant anatomical structures during PLNDGO in order to estimate the actual incidence of complications.

The changes are: Text on page 24. Lines: 651 - 666

Thank you very much for your valuable input and suggestions for the manuscript.

Thank you,

Sincerely

Stoyan Georgiev Kostov, MD

Clinic of Gynecology

MHAT “Saint Anna”   ,Street “Car Osvoboditel” 100

Varna 9000, Bulgaria

Tel: +3590896711178
